# Psychological Factors that Lessen the Impact of COVID-19 on the Self-Employment Intention of Business Administration and Economics’ Students from Latin America

**DOI:** 10.3390/ijerph17155293

**Published:** 2020-07-22

**Authors:** Brizeida Raquel Hernández-Sánchez, Giuseppina Maria Cardella, José Carlos Sánchez-García

**Affiliations:** Department of Social Psychology and Anthropology, University of Salamanca, 37005 Salamanca, Spain; brizeida@usal.es (B.R.H.-S.); jsanchez@usal.es (J.C.S.-G.)

**Keywords:** Covid-19, psychological need satisfaction, optimism, proactiveness, well-being, mental health, entrepreneurial intention

## Abstract

Background: The 2019 coronavirus disease epidemic (Covid-19) is a public health emergency of international concern and poses a challenge to the labor market. The pandemic has a devastating and disproportionate effect on young workers, their interest in entrepreneurship, and their mental health. Research is needed to develop evidence-based strategies to improve coping and reduce adverse psychological problems. The objective of this study was to analyze the impact that Covid-19 pandemic perception and psychological need satisfaction have on university students and their self-employment intention. In addition, we also analyzed the role of moderation played by psychological aspects. These psychological factors (i.e., Optimism and Proactiveness) can also improve young people’s mental health and well-being. Methods: An explorative study (online survey) was conducted in March 2020 934 university students from Latin America. Regression analysis models were built to examine the relationships between Covid-19 pandemic perception, personality variables, and entrepreneurial intention. Mediation models, through the bootstrapping method, were performed to analyze the mediating role of proactiveness and optimism. Results: Results indicate that students’ perception of Covid-19 and psychological need satisfaction are associated with entrepreneurial intention. Additionally, the present study argues that proactiveness and optimism mediate these relationships. Conclusions: This study identifies psychological factors associated with a lower level of Covid-19 impact and that can be used for psychological interventions that result in an improvement in the mental health of these vulnerable groups during and after the Covid-19 pandemic. Theoretical and practical implications are discussed.

## 1. Introduction

Entrepreneurship is a determining factor for economic and social development [1,2], enhances the creation of wealth and value [3], and improves the well-being in nations [4]. Over the years, entrepreneurship has received a broad consensus in the academic and international communities on its importance, however, research on its relevance in uncertain and adverse situations is rather scarce [5].

Covid-19 (coronavirus disease 2019) is a serious disease caused by a new type of coronavirus (SARS-CoV-2) first detected in Wuhan, China in December 2019. The virus has spread rapidly throughout the world [6], registering more than 5,000,000 cases and more than 300,000 deaths. On 11 March 2020, with 114 countries involved, the WHO officially decreed a state of pandemic [7].

The Covid-19 pandemic is a violent shock both from a health-related point of view and also for the global economy, generating an unprecedented environment of high uncertainty [8].

States have already taken a series of budgetary and liquidity policy measures to increase the capacity of their health and economic systems, and to support particularly affected citizens and sectors, however, the outlook remains bleak.

For example, the coronavirus pandemic has a strong impact on physical and psychological health [9]. In China, 58.3% of subjects feel psychologically involved, showing a decrease in positive feelings and satisfaction with life [10].

This is echoed by the economic problems that can help influence people’s quality of life. Social and economic factors are the drivers of the conditions in which people live. Employment, safety, income, education, social support, and discrimination factors account for around 40% of all health [11].

The idea that economic and social factors are related to the health and satisfaction of the individual provides a starting point for an economy that is more focused on human well-being.

Recent studies have considered well-being as the result of individuals’ participation in the social, economic, political, and cultural processes of the community [12,13,14]. In an attempt to interact with people’s lives, economic relationships are understood as integrated into a broader context. An approach in line with Polanyi’s notion of economics as an established process [15].

According to the European Commission survey published on 6 May 2020, the economies of 72 member countries are expected to experience a 7.5% loss in 2020 and grow by about 6% in 2021. The shock to the EU economy is symmetric in that the pandemic has hit all member states, but both the drop in output in 2020 and the strength of the rebound in 2021 are set to differ markedly. Each Member State’s economic recovery will depend not only on the evolution of the pandemic in that country but also on the structure of their economies and their capacity to respond with stabilizing policies. Furthermore, consensus estimates for the initial unemployment claims in the US were around 1.6 million in May, but the figure reached 3.28 million, a historically unprecedented figure, approximately five times greater than the largest weekly increase in the global financial crisis [16].

In China, the Covid-19 pandemic has had an impact mainly on Small and Medium Enterprises (SMEs). In February 2020, 30% of companies saw their revenues fall by more than 50%, while a third saw a decrease of 20–50%. Estimates also indicate that every ten days of job loss in the Chinese economy reduces quarterly GDP growth from 0.39% to 0.46% [17].

Latin America, used to facing negative external shocks, is today one of the areas most affected by the Covid-19 pandemic from an economic perspective. According to the Economic Commission for Latin America and the Caribbean (ECLAC), the economic performance registered in 2019 was poor, with a GDP growth of only 0.1%. Before the pandemic outbreak, the expected expansion for the region in 2020 was 1.3%, mainly driven by the recovery of the two main economies of the continent: Brazil and Mexico. Although it is extremely difficult to estimate the impact of the coronavirus on growth, several analysts argue that a recession of −2.5% to −5.2% in 2020 is a plausible outcome [18]. To this is added another problem: in Latin America, approximately 130 million workers (53% of the employed population) do so informally. High levels of informal work represent a great challenge.

According to Brück, Llussá, and Tavares [19], events such as wars, natural disasters, terrorist attacks, and pandemics have the potential to influence the expectations and perceptions of the population of entire countries and are generally associated with a decrease in investment and GDP, as well higher levels of uncertainty. Furthermore, they present challenges not only for existing companies but also for the creation of new businesses and commercial activities in general [20].

Given this reality that we are living with the Covid-19 pandemic, and which strongly influences our health, economy, and society, we ask ourselves how it will affect entrepreneurial development, specifically how it will affect potential entrepreneurs, our potential business creators. In the literature, the entrepreneurial intention has been considered the most immediate antecedent of business creation [21,22,23,24,25]. The entrepreneurial intention has been defined as a mental state that focuses the attention of individuals on performing certain behaviors related to business creation [26,27]. It is not surprising that, due to their crucial role as a preliminary stage in the business creation process, researchers have focused their efforts on researching the factors that positively influence people’s intention to start a business [28,29].

Despite the considerable contribution of the scientific literature on the processes that favor entrepreneurship, personality factors, and their relationship with the intention to start a business have not received enough attention in situations of uncertainty, catastrophes, pandemics, etc. [30,31,32,33].

Psychological need satisfaction, in previous researches, has been considered an interesting intrinsic motivational factor for understanding the business process [34] and that could act as a driver to face a crisis. The satisfaction of basic psychological needs has a positive effect on the well-being and growth of the individual, while the lack of satisfaction translates into low motivation and a feeling of discomfort that can even lead to the onset of mental illness. As many academics point out [35,36], the satisfaction (or lack thereof) of basic psychological needs depends on the environment in which the subject lives. In general, supportive environments promote well-being in a subject, while frustrating environments generate malfunction and discomfort [37].

Studies have analyzed these relationships in safe situations. Little or nothing is known of the effect they could have on situations perceived as dangerous. Furthermore, the perception of the pandemic itself could influence individuals’ entrepreneurial intentions through the subjective perception of danger, blocking the positive behaviors, and frustrating the basic psychological needs satisfaction. This is one of the main objectives of this work, which tries to fill this gap in the literature.

In an environment perceived as uncertain and dangerous, the effect of Covid-19 pandemic perception and psychological need satisfaction may depend on additional factors. One of them is proactiveness.

Only in recent years have some studies shown a positive association between proactive personality and entrepreneurial intentions [38,39]. For example, in the study by Hansen et al. [40], proactiveness, which is related to the ability to find opportunities and possible solutions in uncertain situations, was considered one of the main factors that influence entrepreneurial intentions.

Ideologically connected with proactivity, optimism appears to also be considered (especially in adverse situations, such as the current pandemic) a particularly important factor for entrepreneurs to pursue their long-term goals [41,42]. Having a positive perspective on future results, focusing attention on positive stimuli, and avoiding those that are perceived as threatening leads to an increase in entrepreneurial intention and, consequently, predisposes the individual to implement a specific behavior [43] and generated well-being. A study by Kleiman et al., [44] demonstrated the multidimensional nature of optimism which appears to be related to fewer depressive episodes as a consequence of stressful and anxiety-provoking life events.

Given the historical period we are experiencing, characterized by the Covid-19 pandemic, we conducted this study to increase our understanding of the factors that influence entrepreneurial intentions in situations of crisis and uncertainty, examining, in particular, the influence of Covid-19 pandemic perception and psychological need satisfaction in entrepreneurial intentions, as well the possible mediating role of proactiveness and optimism.

Understanding the dynamics of these relationships is particularly important for the numerous governments and companies present in the area facing this economic shock. By understanding what drives entrepreneurial intentions in adverse conditions, we can design better programs to effectively improve business efforts.

To meet these goals, we conducted an explorative study with a sample of college students to examine the effects of Covid-19 pandemic perception, psychological need satisfaction, proactiveness, and optimism are in entrepreneurial intentions. In this work, we first develop the theoretical background of our research. Based on this, we present the hypotheses, which are then empirically tested with the data from a survey carried out in 934 university students. The methods are described in the next section, after which we provide the results of our empirical study. We conclude with a discussion of the main results in relation to the previous literature, as well as suggest some recommendations for future policies and lines of research.

## 2. Literature Review and Hypothesis Development

Theoretical models that analyze the entrepreneurial process [45,46] emphasize the importance of personal, cognitive, and prescriptive factors to determine the probability that an individual would be willing to start a new business. Among the main factors related to entrepreneurial intentions are personality traits associated with entrepreneurs [30,31,33]. The results of previous studies suggest that personality traits substantially influence how entrepreneurs think, the objectives they set, and, through their actions, what they achieve [22,47].

In particular, previous studies have established that an entrepreneur generally has an optimistic orientation towards the future and seems more capable of finding opportunities and achieving the desired objectives [38,48,49].

This research will focus on four characteristics, namely proactiveness, optimism, the Covid-19 pandemic perception, and psychological need satisfaction, to quantify the relationship between these four traits of the students and their entrepreneurial intention. In summary, this approach will allow this research to determine the relative importance of the four business characteristics to predict the EI of students in an adverse situation. Despite literature supporting the link between psychological and personality factors and entrepreneurial intentions, these four characteristics have been ignored in the literature.

The four dimensions are briefly described below.

### 2.1. Entrepreneurial Intention

Entrepreneurial intention has been considered the most important predictor of behavior [46,50,51]. By intention, we can understand “a state of mind directing a person’s attention (and therefore experience and action) towards a specific object (goal) or a path to achieve something (means)” (Bird [52] pp. 442). Intentions are related to a plurality of factors (beliefs, values, habits, needs) which also include cognitive factors, which can improve or decrease these intentions.

Over the years, different models of intentions have been developed in the field of entrepreneurship [45,52,53,54,55]. In general, scholars who have analyzed the gap between intentions and behaviors tend to consistently demonstrate this, as much as 39% of the variance in actual behavior can be explained by intentions [56,57,58]. Therefore, more than half of the variance remains unexplained. Several studies [52,56,58] have tried to remedy this lack, through the inclusion of some factors and moderators additional.

Some studies indicate that other predictors, on a personal or social level, may influence entrepreneurial intentions. For example, according to Krueger et al., [54] the intentions are developed from the perceptions of desirability and viability of the entrepreneurial action. After this, a situation perceived as uncertain or dangerous can influence a person’s intention to start a business. For example, considering the current pandemic situation, the perception of the lack of safety in travel, for commercial reasons, combined with the fact that many aspects of public and private life have become online [59], could have a negative impact on the viability of behavior. Krueger et al., [54] also argue that self-efficacy positively influences the viability of entrepreneurship and that the personal propensity to seek opportunities and act on these opportunities (proactiveness in this study) can facilitate the development of intentions. As proactivity is closely linked to identification opportunities, it plays an important role in encouraging new businesses.

In other words, in the case of contexts perceived as dangerous or adverse, other variables associated with the individual’s psychological, cognitive, and personality traits can still help develop entrepreneurial intentions.

Some studies on entrepreneurial intentions [60,61,62,63,64] have laid the foundations for a deeper examination of the intentions of people in situations of insecurity. Specifically, the strength of the intention to become a self-employed person would depend on the tolerance that the subjects show for the risk and on their preference for independence.

### 2.2. Covid-19 Pandemic Perception

According to the approach of social-cognitive theory [65], the surrounding environment influences the behavior of the individual through personal perceptions. Not all people perceive the same situation with the same degree of severity. Taking these differences into account and understanding them is important to analyze behavioral responses, especially in adverse situations.

In the literature, few studies have focused on perceptions derived from an unsafe, dangerous, and risky environment, and the impact these perceptions can have on the intention of starting a business. For example, Gaibulloev and Sandler [66] suggest that terrorism and other violent events could hinder business growth by increasing business costs that reduce profits and returns, discouraging the creation of new businesses.

Among the few studies relating to dangerous situations, we find those related to the scenarios of war [67,68], terrorism [69], and natural disasters [70,71]. Currently, there is no empirical evidence that has analyzed the psychological and cognitive factors that could influence entrepreneurial intention in relation to a pandemic situation. Current studies have focused more on the general economic aspect [17,72].

To describe the literature on the subject, we will refer explicitly to the more general concept of perception of danger. Previous studies have shown the negative relationship between perception of danger and intention for business. Bullough, Renko, and Myatt [67] examined the effects of perceived danger, self-efficacy, and resistance on business intentions under unfavorable conditions during the war in Afghanistan. Their results suggest that perceived danger was negatively related to an individual’s entrepreneurial intentions but marginally less among highly resistant individuals. Jahanshahi, Zhang, and Gholami [73] also in Afghanistan, studied the antecedents of the persistence of companies run by female entrepreneurs. The results showed that female entrepreneurs with a high level of internal locus of control have higher levels of resilience. Furthermore, the perceived danger and influence of supporting family businesses (the first negatively, the second positively) resulted in the persistence of female-led businesses.

These studies highlight the importance of the interaction of psychological, social, and environmental factors in shaping the entrepreneurial capacity of people in uncertain situations.

### 2.3. Psychological Need Satisfaction

Motivational factors have proven to be an important paradigm for the study of the determinants of an entrepreneurial career choice. On the one hand, personal motivational factors act to seek, through business, a possible form of self-realization and to escape the unemployment trap. On the other hand, there is a strong and profound interaction between family, social and institutional factors, within which the entrepreneurial career choice appears as a “response” linked to the satisfaction of basic psychological needs.

According to Self-Determination Theory (SDT) [74,75], motivation is a vital source that makes people persistent and active in implementing their behaviors [76]. Specifically, SDT classifies motivation on a continuum, ranging from extrinsic motivation to intrinsic motivation. What leads to intrinsic motivation is the satisfaction of three basic psychological needs conceptualized by SDT: autonomy, competence, and relatedness. It is argued that these needs are necessary for the well-being and persistence of behavior [77] and that they play a role in defining aspirations and, therefore, in professional choices [78]. When these needs are not met, intrinsic motivation will be hampered, which is called psychological need frustration [76,79] and, as a result, people could develop health problems and persist less in purposeful behaviors. Conversely, if these needs are met, they are more likely to engage persistently in entrepreneurial behavior [76]. In this case, we speak of the psychological need satisfaction, which is the factor we will refer to in the present study.

The perception of autonomy, of being competent, and of having a sense of closeness with others, is at the heart of entrepreneurial research, but, surprisingly, this has rarely been explored in previous studies on entrepreneurship, meaning the mechanisms behind motivations have been ignored [80]. However, the few studies carried out have highlighted its indisputable relevance. For example, Baluku et al., [81] found that the need for autonomy satisfaction is positively related to Entrepreneurial Intention (EI) and that business mentoring is related to EI in individuals who show higher levels of autonomy.

In an interesting study with university students in Yemen, Al-Jubari [82] tested the influence of satisfying the basic psychological needs of SDT on business intention through the factors of the theory of planned behavior. The results of the study support the theoretical integration of the model, in which the satisfaction of basic psychological needs has a positive effect on the attitude towards entrepreneurship, subjective norms, and perceived behavioral control and these, in turn, have a positive effect in the entrepreneurial intention.

### 2.4. Proactiveness

Bateman and Crant [83] discussed the importance of proactiveness in organizational behavior, defining it as the ability to achieve significant change in the environment by identifying opportunities. In reality, it is not just about important attributes of flexibility and adaptability towards an uncertain future. Being proactive is taking the initiative in improving business as well. For years, researchers have debated the nature of proactiveness. According to some scholars, it is a stable disposition of the individual [84], others have considered it a model of general behavior [85], and others, lastly, a specific attitude in the workplace [86].

In general, the proactive approach considers the possibility that individuals create their own environment, that behavior is controlled by factors internal and external to the individual, and that situations are a function of people and vice versa [87]. There are mutual causal links between the person, the environment, and the behavior [88], therefore, individuals can intentionally and directly change their current circumstances and given situations.

This proactive orientation has been discussed in the entrepreneurial process. Much research has found a significant relationship between proactive personality and entrepreneurial behavior [89,90,91], even during the economic crisis [92]. Shapero and Sokol [45] discussed a trend towards action and initiative to describe the social dimensions of business events. Krueger and Brazeal [93] included the concept of “propensity to act” in their study of business intentions and potential. Furthermore, Becherer and Maurer [94] related proactiveness with the decision to start a business, as well as with the legacy of a company.

Proactive personality has been linked to professional success [95,96] and as an employability asset [97,98]. Crant et al., [99] in the bibliographic review carried out on proactiveness in the organizational field, provide empirical evidence on a wide spectrum of favorable results linked to higher performance and innovation, emotional commitment, and job satisfaction.

Bell [100] in a study conducted with UK university students, showed that proactiveness, along with risk, are the key factors in positively influencing entrepreneurial intentions. Kumar and Shukla [101] have also achieved similar results. These studies confirm the importance that personality traits, such as proactiveness, have in the entrepreneurial intention.

### 2.5. Optimism

Optimism has been studied little in relation to entrepreneurial intentions. The few studies in the literature have suggested that optimism may play an important role in the entrepreneurial process [41,42,102]. In a crisis, optimistic entrepreneurs believe more in the success of their actions and, therefore, may be more likely to establish a new business [103]. Some have concluded that optimism is a key requirement for entrepreneurship [104,105].

Furthermore, as Krueger & Day [48] states: “Some of the most promising recent models of entrepreneurship focus on cognitive processes, showing the importance of an opportunity-friendly cognitive infrastructure” (pp. 324). For example, Lee et al., [106] discovered that optimism is associated with self-confidence among students. Overall, positive attitudes toward entrepreneurship, such as a possible career choice, and confidence in one’s skills were found to be significantly related to entrepreneurial intentions. A few years earlier, Giacomin, Janssen, and Shinnar [107] found that the students who most expected positive results in entrepreneurship showed higher levels of business intentions. Furthermore, in general, employers have a greater tendency to be more optimistic than other workers [103,108]. However, as far as we know, only the studies by Bernoster et al. [109] and Madar et al. [105] documented a positive relationship between optimism and entrepreneurial intention among students.

In summary, despite the theoretically positive relationship suggested by the literature between optimism and entrepreneurial tendencies, little attention has been paid to the empirical investigation of the relationships between the two constructs [110].

## 3. Hypothesis

As discussed in the previous sections, this study is the first attempt to analyze entrepreneurial intentions in a pandemic situation.

First, through a representative sample of Latin American university students, this study aims to analyze the relationships between proactiveness, optimism, Covid-19 pandemic perception, psychological need satisfaction, and entrepreneurial intention in an adverse situation, as it is this current pandemic. Second, the current study focuses primarily on how the Covid-19 pandemic perception and need satisfaction mediate the predictive effects of student proactivity and optimism on entrepreneurial intention. Consequently, this study raises the following hypotheses:

**Hypothesis 1:** *The Covid-19 pandemic perception are negatively associated with entrepreneurial intentions (H1a) and Psychological Need Satisfaction are positively associated with entrepreneurial intentions (H1b)*.

**Hypothesis 2:** *Proactiveness is positively related to entrepreneurial intentions*.

**Hypothesis 3:** *Optimism is positively related to entrepreneurial intentions*.

**Hypothesis 4:** *The relationship between Covid-19 pandemic perception and entrepreneurial intentions is mediated by the Proactiveness (H4a) and Optimism (H4b)*.

**Hypothesis 5:** *The relationship between Psychological Need Satisfaction and entrepreneurial intentions is mediated by the Proactiveness (H5a) and Optimism (H5b)*.

These hypotheses are represented in Figure 1.

## 4. Materials and Methods

### 4.1. Sample and Procedure

To recruit participants, we contacted several professors from various universities in Latin America, to motivate their students and involve them in completing the questionnaire.

Our sample includes 934 university students from Latin American countries (45.6% from Ecuador, 45.3% from Panama and 9% from other countries), of whom 67.5% are female, and 32.5% male. Regarding the area of knowledge, most of the students are enrolled in the Faculty of Business and Administration (55.4%) and the Faculty of Economic Sciences (23.7%). The remaining 25% is enrolled in the faculty of Social and Health Sciences. The age range is between 18 and 69 years, with an average of 23.6 (SD = 5.94). With previous permission and authorization from authorities, the students were informed about the purpose of the study. The questionnaires were administered online, participation was voluntary, and the data were processed anonymously and confidentially following the ethical criteria established by the A.P.A. (American Psychological Association).

Considering the data collection method (online questionnaire), the only possible method due to the pandemic, the researchers were unable to check the characteristics of the sample, therefore it is not representative of the population studied (Latin America).

### 4.2. Measures

Self-report questionnaires were used to collect data on the study variables. The responses on all scales followed a 5-point Likert format ranging from strongly disagree (1) to strongly agree (5).

To measure Covid-19 pandemic perception, we created an ad hoc questionnaire. The scale consisted of 14 items: 7 items measure the impact of Covid-19 in the country, and the other 7 items measure the impact of the pandemic on the person itself surveyed. The country’s scale did not show good internal consistency, probably because many of the students interviewed live far from their homelands. For this reason, we decided to use only the personal subscale. Examples of items were: “The Covid-19 negatively affects my future” or “The Covid-19 will decrease my job opportunities”. The scale showed good validity with a Cronbach’s alpha value of 0.884.

Psychological need satisfaction was measured using the version developed and validated by Chen et al., [35]. The scale has been translated into several languages, including Spanish. The entire scale is made up of 24 items, 12 to measure need satisfaction, and another 12 to measure need frustration. For our study, we decided to use the 12 items corresponding to the need satisfaction. Examples of some of these items are: “I feel free to choose the things I do”, “I feel connected to the people who care for me and who are important to me”. The scale has a Cronbach alpha value of 0.797.

Entrepreneurial intention was measured with the six-item scale of the Entrepreneurial Orientation Questionnaire (EOQ; COE in Spanish) [111]. Some items on this scale are: “I will make any effort to start and develop my own firm”; “I have thought very seriously about creating a firm.” The scale showed a Cronbach alpha value of 0.926.

To measure proactive personality, the corresponding scale of the Entrepreneurial Orientation Questionnaire (EOQ; COE in Spanish) was used [112]. This scale is made up of 10 items and measures the tendency of the respondents to exhibit proactive behavior. The construction of this subscale was based on the work of Seibert et al. [95,113]. Some items on this scale are: “I am constantly looking for new ways to improve my life”, “If I believe in an idea, no obstacle will prevent me from making it come true”. The Cronbach alpha value for the scale was 0.874.

Finally, to measure optimism, we used the optimism scale of the PROE questionnaire [113]. The scale consists of 9 Likert-type items and measures the tendency of an individual to have positive expectations about the future. Examples of the items that make up the scale are: “I see the positive aspects of things”; “I think I will achieve the main goals of my life.” The scale showed good reliability (Cronbach’s alpha of 0.858).

In line with previous studies on entrepreneurship [114], data were collected on demographic variables such as age, sex, country, and faculty department.

### 4.3. Data Analysis

Data were analyzed using SPSS version 23 (IMB Corp., Armonk, NY, USA) and Amos version 23 (IBM Corp., Armonk, NY, USA). First, descriptive statistics were produced using standard means and deviations for all variables. Correlations between variables were evaluated using Pearson’s correlations. Hierarchical regression analyzes were performed to analyze the impact of the variables on entrepreneurial intention.

To examine the indirect effect of Covid-19 pandemic perception and Psychological Need Satisfaction on entrepreneurial intention through proactiveness and optimism, we used the bootstrap method. The key principle underlining the bootstrap procedure is that it allows the researcher to simulate repeated subsamples from an original database, allowing the stability of the parameter estimates to be evaluated and their values to be reported with a higher degree of precision. Bootstrap evaluates the indirect effect in each data set and establishes confidence intervals for each indirect effect [115].

In this study, we used the following indices to evaluate model fit: the comparative fit index (CFI) and the Bollen fit index (IFI), both with adequate values greater than 0.90 and the root mean square error of approximation (RMSEA) which must have a value of less than 0.08 [116,117]. The level of significance (*p*-value) will be 5%.

The acceptability of the measurement model was assessed by the reliability of individual items, internal consistency between items, the model’s convergent, and discriminant validity. The literature suggests 0.7 as the acceptable value for Cronbach’s Alpha. Average Variance Extracted (AVE) is higher than 0.5 but we accepted 0.4. According to Fornell and Larcker (1981) [118], if AVE is less than 0.5, but composite reliability is higher than 0.6, the convergent validity of the construct is still adequate.

For discriminant validity, the analysis can be performed by the square root of the AVE value. When the square root of the mean variance extraction rate (AVE value) of each measurement question is greater than the correlation coefficient between the variables, it indicates that there is a strong discriminant coefficient between the variables, that is, the difference between each measurement variable is better (see Table A1 and Table A2).

## 5. Results

Before testing the hypotheses, the mean, standard deviations (SD), and correlations between the variables were calculated (Table 1). The average scores indicate that the students in our sample show medium to high levels of optimism, proactiveness, psychological need satisfaction, and entrepreneurial intention. The Covid-19 pandemic perception variable obtained the lowest score (M = 3.05, SD = 0.904).

The dependent variable, entrepreneurial intention, was significantly and negatively correlated with Covid-19 pandemic perception (*r* = −0.24, *p* < 0.01), suggesting that high levels of pandemic perception negatively affect its career choice. Psychological Need Satisfaction (*r* = 0.39, *p* < 0.01), Proactiveness (*r* = 0.45, *p* < 0.01) and optimism (*r* = 0.44, *p* < 0.01) were positively and significantly related to entrepreneurial intentions. These results give us initial support of our hypotheses.

Asymmetry and kurtosis were calculated under the assumption of normal data distribution. All univariate asymmetry values varying between 0.367 and −2.04, while kurtosis values, which must vary between −0.063 y 4.70, thus, meeting with the univariate normality criterion [119]. In accordance with “For sample sizes greater than 300, depend on the absolute values of skewness and kurtosis without considering z-values”. Either an absolute skew value larger than 2 or an absolute kurtosis (proper) larger than 7 may be used as reference values for determining substantial non-normality” (Hae-Young [119] p. 53).

Given that the variables of our study show significant correlations, variance inflation factors (VIF) were calculated to investigate multicollinearity. All VIF values were well below the threshold of 2.5 [120]. Furthermore, none of the correlations exceeded 0.7 [121].

We can, therefore, conclude that multicollinearity is not an issue in this analysis.

The next step of data analysis was to test the goodness-of-fit of the model, using the AMOS Graphics 23.0 software. The fit indices for the proposed model were: χ^2^ = 2,763,073, df = 806, *p* < 0.01, CFI = 0.90, IFI = 0.90, RMSEA = 0.05. The confirmatory factorial analysis showed an adequate adaptation of the data to the model. Although the chi-square was significant (χ^2^ (806) = 2,763,073, *p* < 0.01), due to the large sample size, the other adaptation measurements made confirmed good compatibility.

To examine whether the variables in our study predicted entrepreneurial intention in a statistically significant way, a hierarchical regression analysis was performed (Table 2). The first model includes the control variables: gender, age, country of origin, and faculty department. All variables had a significant effect on business intentions. In model 2, we added the independent variables. As the results show, Covid-19 pandemic perception (β = –0.013, *p* < 0.01) is negatively related to entrepreneurial intentions and Psychological need satisfaction, (β = 0.35, *p* < 0.01) is positively related to entrepreneurial intentions, which supports hypothesis 1. Furthermore, the relation of intention with Proactiveness (Model 3) was positive (β = 0.29, *p* < 0.01), providing support for hypothesis 2. Lastly, model 4 supported hypothesis 3. Optimism was positively related to entrepreneurial intentions (β = 0.16, *p* < 0.01).

To calculate the mediation effect by the Proactiveness and Optimism (H4 and H5), the bootstrap method was used according to the recommendations of Preacher and Hayes [115]. During the mediation test, bootstrap was performed with 5000 iterations and the bias-corrected confidence interval was adjusted to 95%. If the 95% confidence interval does not include 0, then the mediation effect is considered statistically significant at the level = 0.05. Table 3 presents the relationship between Covid-19 pandemic perception and entrepreneurial intentions fully mediated by Proactiveness and Optimism. This is indicated by a significant total effect, which is the sum of the direct and indirect effects (β = −0.21; C.I. [−0.276; −0.162]). At the same time, the specific indirect effects also seem significant (through Proactiveness: β = −0.04; CI [−0.077; −0.026]; through Optimism: β = −0.06; C.I. [−0.091; −0.035]), as well as the direct effect of Covid-19 pandemic perception on entrepreneurial intentions (β = −0.12; CI [−0.179; −0.076]). For this reason, the H4 hypothesis was confirmed.

For hypothesis H5 (Table 4), we confirm that the relationship between Psychological Need Satisfaction and entrepreneurial intentions is mediated by the paths Proactivity (β = 0.19; CI [0.124; 0.280]) and Optimism (β = 0.17; CI [0.096; 0.272]). Again, both the total effect (β = 0.58; C.I. [0.495; 0.669]) and the direct effect (β = 0.27; C.I. [0.178; 0.371]) are positive and significant. We can conclude that hypothesis 5 was confirmed.

## 6. Discussion

Understanding what factors contribute to support entrepreneurship as a career choice in a crisis seems to be of fundamental importance as it translates into the desire for growth and resilience that people can face when faced with an adverse situation.

Specifically, we hypothesized that proactiveness and optimism would mediate negative relationships between Covid-19 pandemic perception and intention, and the positive relationship between Psychological Need Satisfaction and entrepreneurial intention. Our results support the mediation hypotheses and suggest that optimism and proactiveness are particularly important in the entrepreneurial process. Furthermore, we found that Psychological Need Satisfaction supports entrepreneurial intention, but also that the Covid-19 pandemic perception hinders them.

Our model, and the results obtained, are a first attempt to close the gap in our knowledge of what drives entrepreneurship in highly adverse conditions (for example, in economies subject to pandemics).

Our suggested mediation model expands on previous literature, which is mainly focused on the direct relationship between personality and intentions [38]. In particular, the current study joins the few studies in the literature about possible mediation processes that influence the relationship between personality and entrepreneurial behavior [105,122] in situations perceived as negative and high in uncertainty.

In addition to its positive and direct effect on intentions, proactiveness, and optimism interact with the pandemic’s perception and with psychological need satisfaction. Furthermore, and this is an important contribution of this study, optimism and proactiveness attenuate the negative relationship between Covid-19 perception and intentions. Optimism and proactiveness significantly strengthen the positive relationships with entrepreneurial intention, indicating that both traits are important to understand entrepreneurship given dangerous situations such as a pandemic scenario.

As such, according to this study, influencing the entrepreneurial intention of students (potential entrepreneurs), who are hampered by the pandemic situation, means analyzing the psychological and social factors that influence intentions and understanding how they relate to entrepreneurial behavior in the practice. Previous studies have demonstrated a positive influence of proactive personality on entrepreneurial intentions [89]. Therefore, it is highly important to appreciate proactive traits in students so that they can think of entrepreneurship as a possible career option.

We have also shown that optimism is significantly and positively related to entrepreneurial intentions, specifically in difficult times. This importance of optimism for the development of entrepreneurial intentions complements previous studies in which entrepreneurs have been considered individuals who remain optimistic and persist even in adverse situations [123,124].

However, it should be noted in this regard that studies have not always produced consistent results in the literature [108,125]. While some studies have found a positive relationship between optimism and entrepreneurial orientation [108], others have found a negative one [126]. These latest studies have concluded that excessive optimism may be a factor behind wrongful decisions with high levels of risk-taking. However, the conclusions of James and Gudmundsson [127], in line with our study, suggest that moderate levels of optimism have a positive impact on the entrepreneurial process and, consequently, increase the chances of success for a new company.

Furthermore, our results are in line with psychological theories that suggest having an optimistic orientation (positive psychology) and an ability to find alternative solutions are valuable psychological resources that can help deal with stress and facilitate coping strategies [128], especially during crises.

An environment that is perceived as highly dangerous and uncertain is negatively related to the intentions for starting a business of its people. However, the individual perception of a negative situation differs between individuals, with different consequences for their intentions [54].

Certainly, beyond individual specificities, what must be taken into account is the role that the variables of our study have in entrepreneurial intentions.

Previous research has described proactiveness as a resource that people can mobilize in a time of stress or adversity, which allows them to overcome the barriers of various areas of life [129,130]. It seems that having optimistic thoughts about the future is an important factor, especially in uncertain situations. It appears that these effects may have a similar role in the entrepreneurial spirit, allowing the development of entrepreneurial intention in environments of uncertainty.

This study presents an innovative approach because it examines the psychological aspects of entrepreneurial intention in a pandemic situation. Furthermore, our work confirms previous studies that also show the positive role of need satisfaction, which acts as an accelerator of an entrepreneurial initiative by increasing entrepreneurial intention. In this way, it joins the growing literature that explores how basic psychological needs act as motivational drives that positively influence one’s future, including career choice [34]. Our findings have shown that satisfaction of needs is positively related to entrepreneurial intentions in support of the consensus that, to be successful, an entrepreneur must express and use various skills, including those aimed at satisfying his needs and at achieving your own well-being in general [82,131].

Our results come from a sample of Latin American students who are experiencing the consequences of the pandemic, and for whom decisions about ways to rethink their future, including work and entrepreneurship, are important. The entrepreneurial intention in our sample is high. This result is not entirely surprising, both, because of the specificity of the sample composed mainly of students from the business and management and economics fields, but also because of the high rates of youth unemployment that affect the areas of Latin America, which stands at 14% [132], and for which job opportunities seem limited and entrepreneurship represents a solution capable of generating higher economic returns than alternative job opportunities. According to the Multilateral Investment Fund (a member of the Inter-American Development Bank), small and medium-sized enterprises (SMEs) are responsible for 66% of jobs worldwide, on average [133].

In general, the stronger the ability to cope with adverse situations and the greater their optimistic orientation towards the future, their proactiveness and the satisfaction of their basic needs, the greater the probability that people will successfully face a negative event or moment of crisis as a learning experience [134], with positive repercussions on their psychological health. As competent actors in their future, they can develop intentions to change the status quo of the communities in which they live by undertaking entrepreneurial initiatives.

This study has some limitations. First, in this study, one limitation concerned how the sample was recruited. The fact that we contacted the professors for the recruitment of the participants did not allow us to monitor their characteristics, generating an unrepresentative sample.

In addition, the use of self-report data and a single method of data collection raises questions about the variation of the method. However, multicollinearity was not a problem in our study, so we believe that according to Siemsen et al. [135] should not inflate the meaning of the interaction effects.

Furthermore, data collection at a time like this has created certain problems for the research team. The online administration method dictated the use of scales with few items, focusing the research objectives on analyzing the variables considered more important for hypothesis purposes. This could probably be a limitation since longer scales could have led to more robust results [136].

This study could represent a limitation for those who are specifically interested in analyzing entrepreneurial behavior (not simply intention). However, the link between intention and behavior has been widely documented in the literature [21,52,137]. We believe that future research that also focuses on behavior could help improve our understanding of the intention-behavior relationship, even under critical conditions.

Our study offers important implications for all those who are called to promote entrepreneurship in notoriously difficult situations, such as in economic crises. All governments are making great efforts to face the economic crisis and help countries to overcome the challenges that the pandemic situation has imposed. Institutional support seems important both to preserve the businesses present in the area and to launch new businesses that could work as a solution to the discouragement that the current pandemic is generating. Therefore, the main objective of these adverse conditions should be based on creating a social and institutional environment that potential entrepreneurs can perceive as safe.

Economic growth has raised living standards worldwide. Modern economies have often used the standard metric of economic growth, the Gross Domestic Product (GDP), as a unit to indicate the development of a nation, combining its economy with the well-being of society. As a result, policies leading to economic growth are considered beneficial for society. Furthermore, considering the close relationship between health and economic growth, this study represents a moment of reflection about the psychological factors that can influence people’s lives and their well-being.

When the environment is uncertain, people are likely to draw on their skills and develop entrepreneurial intentions when they believe they have the skills to find alternative solutions to overcome obstacles (to be proactive) by taking advantage of more positive aspects of the situation (to be optimistic). Proactiveness helps people focus on long-term goals and productive activities, even when times are tough. Of course, improving people’s confidence about the future and their proactive behavior does not guarantee that they will be successful as entrepreneurs, it only increases the probability that they will consider an entrepreneurial path.

For example, since both proactiveness and optimism can be encouraged, entrepreneurship study programs should be intensified within schools and universities [138]. For example, educators should work directly training students and promoting curriculum projects that foster the development of useful personal skills to develop an entrepreneurial mindset [139]. These activities are particularly important in developing high standards of personal performance and expectations for performance-based results [140]. Entrepreneurship and education, if interconnected, have extraordinary potential because by developing the human capital necessary to build the society of the future, as they generate employment and economic growth [141].

In addition to the contribution of this study to the importance of the social and institutional sphere; and the acquisition of skills among young people, these results also have important implications at the organizational level. Times of crisis can become an opportunity for enterprises to become more innovative. Faced with outside pressures, the challenge for business leaders is to break out of their comfort zone and routine to become creative problem solvers and rediscover their entrepreneurial spirit. Generating innovation in companies in a constantly changing market is essential for the survival and success of a company. As Tellis [142] suggested, the key to innovation is human capital, therefore, understanding these elements of personality that drive innovation among employees will certainly contribute to the success of businesses.

## 7. Direction for Future Research

Longitudinal research analyzing who has embarked on an entrepreneurial career in situations characterized by uncertainty and adversity is an important direction for future research, especially given the importance of entrepreneurship for the economic and sustainable development of a community.

In this study, we analyze the perceived effects of the Covid-19 pandemic in a specific context (Latin America). Naturally, the perception of the pandemic can be felt differently in many other parts of the world. An interesting path for future research could include analyzing several nations so that the results can be compared with countries with different economies, as well as with countries that are considered entrepreneurial-friendly. Such research could address issues related to the importance of proactiveness and optimism based on the level of perceived adversity, and if whether the most proactive and optimistic entrepreneurs are the ones who are best able to overcome the challenges posed by the pandemic.

## 8. Conclusions

This study is particularly important in the current moment of international economic crisis as a consequence of the Covid-19 pandemic as it involves the production structures of many countries, which are at risk of an implosion in terms of economic growth, the most evident effects of which they can already be seen in the processes of a reduction in the labor market of numerous employees, and by the increasing phenomena of discouragement and/or resignation from job participation, especially in younger generations.

Our study found that the perception that university students have of the Covid-19 pandemic is decreasing their intentions to start a business, with repercussions on their psychological needs. How can we solve this? In our study, we have considered two personality traits that manifest themselves as keys to enhancing the intention to undertake in this specific situation: proactiveness and optimism. These traits can dampen the effect of Covid-19’s negative perception on the intention to start a business, and also enhance the well-being and mental health of these young people to be able to self-employ.

## Figures and Tables

**Figure 1 ijerph-17-05293-f001:**
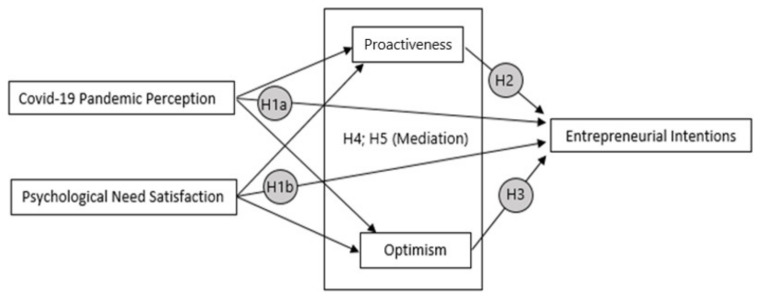
Conceptual model.

**Table 1 ijerph-17-05293-t001:** Means, standard deviations, and correlations among variables (N = 934).

Variables	Mean	SD	1	2	3	4	5	6	7	8	9
1. EI	4.10	0.822	1								
2. Covid-19 Pandemic Perc	3.05	0.904	−0.24 **	1							
3. Psic. Need Satisf.	4.39	0.558	0.39 **	−0.21 **	1						
4. Proactiveness	4.00	0.510	0.45 **	−0.16 **	0.39 **	1					
5. Optimism	4.09	0.567	0.44 **	−0.23 **	0.51 **	0.65 **	1				
6. Gender	1.68	0.469	−0.05	0	0.03	−0.07 *	−0.02	1			
7. Age	23.62	5.94	0.06 *	−0.02	0.09 **	0.06 *	0.14 **	−0.04	1		
8. Countries	2.16	2.28	0.08 **	−0.12 **	−0.02	0.06	0.03	0.06 *	0.06	1	
9. Faculties	2.61	1.63	−0.03	−0.01	0.01	−0.01	0.01	−0.07 *	0.02	−0.03	1

* *p* < 0.05; ** *p* < 0.01.

**Table 2 ijerph-17-05293-t002:** Results of regression models—dependent variable = entrepreneurial intention.

Variables	Model 1 (Control)	Model 2 (Hypothesis 1)	Model 3 (Hypothesis 2)	Model 4 (Hypothesis 3)
Gender	−0.06 *	−0.07 **	−0.05	−0.05
Age	0.09 **	0.05 *	0.04	0.02
Countries	0.011 **	0.10 **	0.07 **	0.07 **
Faculty	−0.25 **	−0.20 **	−0.17 **	−0.17 **
Covid-19 Pandemic Perception		−0.13 **	−0.11 **	−0.09 **
Psychological Need Satisfaction		0.35 **	0.22 **	0.18 **
Proactiveness			0.29 **	0.20 **
Optimism				0.16 **
R-square	0.076	0.235	0.302	0.315
Adjusted R-square	0.072	0.230	0.296	0.309
Std. Error	0.79	0.72	0.68	0.68
F-value	18.88 **	47.17 **	56.84 **	52.95 **

** *p* < 0.01; * *p* < 0.05. Beta coefficients are standardized.

**Table 3 ijerph-17-05293-t003:** Total effect, indirect effect, and direct effect Covid-19 pandemic perception—entrepreneurial intentions.

Effect	β	S.E.	Bootstrapping C.I. (95%)	
			Lower Bounds	Upper Bounds
Total Effect	−0.21	0.02	−0.276	−0.162
Covid-19 pandemic perception—Proactiveness—EI	−0.04	0.01	−0.077	−0.026
Covid-19 pandemic perception—Optimism—EI	−0.06	0.01	−0.091	−0.035
Indirect Effect	−0.11	0.01	−0.148	−0.074
Direct Effect	−0.12	0.02	−0.179	−0.076

Beta coefficients are standardized. S.E. = Standardized Error.

**Table 4 ijerph-17-05293-t004:** Total effect, indirect effect, and direct effect psychological need satisfaction—entrepreneurial intentions.

Effect	β	S.E.	Bootstrapping C.I. (95%)	
			Lower Bounds	Upper Bounds
Total Effect	0.58	0.04	0.495	0.669
Ps. Need Satisfaction—Proactiveness—EI	0.19	0.03	0.124	0.280
Ps. Need Satisfaction—Optimism—EI	0.17	0.04	0.096	0.272
Indirect Effect	0.37	0.05	0.279	0.482
Direct Effect	0.27	0.04	0.178	0.371

Beta coefficients are standardized. S.E. = Standardized Error.

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
