# Peer review of "Psychological Factors that Lessen the Impact of COVID-19 on the Self-Employment Intention of Business Administration and Economics’ Students from Latin America"

_ijerph, 2020, doi:10.3390/ijerph17155293_

Round 1

Reviewer 1 Report

The study sample has a strong self-selection bias. Consequently, it does not come as a surprise that you have very high correlation among all study variables. I suggest adding adequate statistical tests to rule out multicollinearity and to also include a confirmatory factor analysis to establish discriminant validity among latent variables.

I suggest to first check for normality (values for skewness and kurtosis) and then to proceed to test the assumption of multicollinearity (Bryman & Cramer, 2005; Field, 2005).

Reviewer 2 Report

The paper deals with a recent and "hot" topics - entrepreneurial intentions among students in the times of covid-19 pandemic. Albeit there are numerous studies on the entrepreneurial intentions, the impact of covid-19 is relativelly unknown. Thus, the paper has a potential to make a significant contribution to the literature on entrepreneurship and entrepreneurial intentions.

My overal assesment of the paper is rather positive, but Authors should make some serious corrections before the paper could be considered for publication.

First, the motivations of the study are not clear. Why students in Latin America and why Ecuador and Panama are so overrepresented, while Brazil, Argentina, Venezuela or Columbia + Mexico are excluded? This should be explained better.

Second, the sample is certainly not representative for entire LA, but also is not representative for Ecuador and Panama! You have run an internet survey, so you cannot control the selection of respondents. You should be quite frank about these limitations both when describing the procedure of data collection, discussing the results and policy implications. Your study is just EXPLORATIVE and this should be underlined in the beginning, maybe even in the title of the paper.

Third, in the discussion you omit important recent references on this topic - your results should be compared inter alia to Udayanan (2019) and Zamrudi & Yulianti (2020).

Fourth, in the abstract you write "the mediating role of proactive and optimism" - IMHO you rather should use term "proactiveness" (noun) or "proactivity" than adjective (proactive), as the former is used widely in the literature (for instance: Brzozowski et. al, 2019)

I wish you good luck in developing your paper!

References:

Zamrudi, Z., & Yulianti, F. (2020). Sculpting Factors of Entrepreneurship among University Students in Indonesia. Entrepreneurial Business and Economics Review, 8(1), 33-49. https://doi.org/10.15678/EBER.2020.080102

Udayanan, P. (2019). The Role of Self-Efficacy and Entrepreneurial Self-Efficacy on the Entrepreneurial Intentions of Graduate Students: A Study among Omani Graduates. Entrepreneurial Business and Economics Review, 7(4), 7-20. https://doi.org/10.15678/EBER.2019.070401

Brzozowski, J., Cucculelli, M., & Peruzzi, V. (2019). Firms’ proactiveness during the crisis: Evidence from European data. Entrepreneurship Research Journal, 9(3), 1-19. https://doi.org/10.1515/erj-2017-0215

Author Response

Please see the attchment

Reviewer 3 Report

Dear editor, 

  It is an interesting article but a number of things need to be improved.   In general English is fine although there are a number of issues throughout the abstract and article with incomprehensible sentences.    There are problems with the title, see comments in the article.    The introduction and part before they start the article at 4. Materials and Methods is way too long and repetitive.    It is unclear how they selected their participants. It's also unclear why there are so many female participants. Is that because the majority of students in business administration etc are women? or is it a form of selection bias?    Is it also not logical that many business administration and economic students want to start their own business and already have certain traits of entrepreneurs? This also means that you cannot generalise to university students in general as you are doing in your title.    The discussion and most part of the conclusion should be grouped together and turned into one discussion. The repetitive parts should be removed.    Limitations and future research should be split up into strengths and weaknesses and future research.    The conclusion needs to be rewritten because at the moment it's discussion part 2 and not a conclusion.    See also comments in the article. 

Round 2

Reviewer 3 Report

There are still a number of little things see comments in the article

Author Response

Thanks for your valuable comment and suggestions to enhance the overall quality our the manuscript. In the revision, we have tried our best to address all the issues raised by you (in the revised manuscript they are underlined in red).

We want to make some considerations.

As for the gap between intentions and behaviors (lines 171-172) you asked us: "Do you mean that they tried to find an explanation for this", the answer is yes, in our manuscript we explain that "In fact, some studies indicate that other predictors, on a personal or social level, may influence entrepreneurial intentions. For example, according to Krueger et al., [54] the intentions are developed from the perceptions of desirability and viability of the entrepreneurial action" (Lines 176-177).

You also suggest adding at least one strength of our study. We believe that in many parts of the manuscript we have developed this point, for example when we write that, considering the few studies in the literature on the entrepreneurship in adverse contexts: "Our model, and the results obtained, are a first attempt to close the gap in our knowledge of what drives entrepreneurship in highly adverse conditions (for example, in economies subject to pandemics)" (Lines 472-474).

Or when we emphasize the fact that: "Our suggested mediation model, in fact, expands on previous literature, which is mainly focused on the direct relationship between personality and intentions [38]. In particular, the current study joins the few studies in the literature about possible mediation processes that influence the relationship between personality and entrepreneurial behavior [105, 123]" (lines 475-478).

We believe that it is a repetition of what has already been stated in the discussion and in some ways in the practical implications that could derive from the present study.

Once again, we appreciate the time you have spent to significantly improve our manuscript. We sincerely hope that our review has adequately addressed your concerns and you will agree that our manuscript can make a sufficient contribution to the literature.

Best regards.